# Carotenoid Contents of *Lycium barbarum*: A Novel QAMS Analyses, Geographical Origins Discriminant Evaluation, and Storage Stability Assessment

**DOI:** 10.3390/molecules26175374

**Published:** 2021-09-03

**Authors:** Ruru Ren, Yanting Li, Huan Chen, Yingli Wang, Lingling Yang, Chao Su, Xiaojun Zhao, Jianyu Chen, Xueqin Ma

**Affiliations:** 1Department of Pharmaceutical Analysis, School of Pharmacy, Key Laboratory of Hui Ethnic Medicine Modernization, Ministry of Education, Ningxia Medical University, 1160 Shenli Street, Yinchuan 750004, China; luckyrenrr@163.com (R.R.); liyanting134@163.com (Y.L.); lovechenhuan520@163.com (H.C.); 18737803095@163.com (Y.W.); 15926361499@163.com (L.Y.); scwsry@163.com (C.S.); zhaoxiaojunwr@163.com (X.Z.); 2Department of Chemical Drugs Analysis, Ningxia Institute of Drug Control, Intersection of Ning’an Street and Feng Yue Alley, Yinchuan 750001, China; 3Department of Pharmacology, College of Pharmacy, Fujian University of Traditional Chinese Medicine, No. 1, Huatuo Road, Minhoushangjie, Fuzhou 350122, China

**Keywords:** *Lycium barbarum* L., carotenoids, QAMS, *trans*-β-Apo-8′-carotenal, storage stability

## Abstract

Given the standard substances of zeaxanthin and its homologues obtained from *Lycium barbarum* L. (LB) are extremely scarce and unstable, a novel quantitative analysis of carotenoids by single marker method, named QAMS, was established. Four carotenoids including lutein, zeaxanthin, β-carotene, and zeaxanthin dipalmitate were determined simultaneously by employing *trans*-β-apo-8′-carotenal, a carotenoid component which did not exist in LB, as standard reference. Meanwhile, β-carotene, another carotenoid constituent which existed in LB, was determined as contrast. The QAMS methods were fully verified and exhibited low standard method difference with the external standard method (ESM), evidenced by the contents of four carotenoids in 34 batches of LB samples determined using ESM and QAMS methods, respectively. HCA, PCA, and OPLS-DA analysis disclosed that LB samples could be clearly differentiated into two groups: one contained LB samples collected from Ningxia and Gansu; the other was from Qinghai, which was directly related to the different geographical location. Once exposed under high humidity (RH 75 ± 5%) at a high temperature (45 ± 5 °C) as compared with ambient temperature (25 ± 5 °C), from day 0 to day 28, zeaxanthin dipalmitate content was significantly decreased, and ultimately, all the decrease rates reached about 80%, regardless of the storage condition. Our results provide a good basis for improving the quality control of LB.

## 1. Introduction

For centuries, *Lycium barbarum* L. (LB) has been widely used for health care in both food and medical fields all over the world, especially in China [1,2]. In recent decades, given its health-promoting properties, LB successfully attracted more peoples’ attention in Europe and North America, where it was known as “super-food” [3]. According to the theory of traditional Chinese medicine (TCM), LB mainly acted on the liver and kidney meridians, and thus could nourish the liver and kidney [4]. Modern pharmacological studies also confirmed that LB possesses excellent activities including anti-aging and antioxidant properties [5,6,7], neuroprotection effects [8], hypoglycemic and hypolipidemic activities [9], and beneficial immune regulation [10].

Carotenoids are one of the most important pigments widely distributed in nature, including the edible medicinal plant LB [11], of which zeaxanthin and its esters are the dominant constituents [12,13]. Among them, β-carotene exhibits similar vitamin A activity [14], and zeaxanthin could suppress age-related macular degeneration [15], scavenge free radicals [16] and reduce the incidence of cardiovascular disease [17]. Due to the excellent bioactivities, several published studies had established the corresponding quantitative methods for the determination of carotenoids existing in LB [3,13,14,18,19], and also supplied a high-speed counter-current chromatography method for obtaining the high purity zeaxanthin dipalmitate [20]. However, zeaxanthin and its ester are extremely unstable which make the standard substances are extremely hard to obtain and store, thus resulting in an extremely high cost. Furthermore, more components in the sample have been determined, and more corresponding authentic standard substances are needed, which greatly limits the practical application of the published methods. Therefore, a stable, cheap and commercially available substitute standard reference was proposed to exert the qualitative and quantitative analysis of multiple carotenoids in LB.

Qualitative and quantitative analysis of multiple components by a single marker (QAMS) is a simple, efficient, and economic method. In this method, only one standard reference was needed, then all analytes in sample could be identified and assayed simultaneously, thus solving the problem of authentic standard substance scarcity [21]. To date, the QAMS method has been applied to the quality control of a series of herbal medicine and foods, and some of them were officially recorded in the Chinese Pharmacopeia, United States Pharmacopoeia, and European Pharmacopoeia [21]. All of the above data proved the reliable and effective of QAMS, which has become a new development trend of quality control of TCM. According to the theory of QAMS, the quantity (mass or concentration) of analyte within a certain range is proportional to the detector response such as peak area or height. By introduction of a relative correction factor (RCF, *f_S/X_*) between analyte and standard reference, all analytes can be assayed simultaneously by using only one standard reference combined with the value of the RCF [22]. Therefore, RCF between reference and analyte is particularly important, which directly influences the accuracy of the quantitation. According to our previous efforts [21,23], we had established several ways to calculate RCF: one was determined by the ratio of the concentrations and HPLC peak areas between the standard reference and analyte, the second was the proportion of the slopes established by chromatographic/spectra calibration curve, and the third was the proportion of the absorption coefficients. Most of the published methods employed one of the main constituents which existed in the sample as standard reference to calculate RCF.

Based on our previous experiments and published data, carotenoids were found in large quantities in LB, and zeaxanthin dipalmitate was the main component. However, zeaxanthin dipalmitate was highly unstable; it was difficult to isolate and obtain the relatively high purity standard reference according to the published data and our efforts. In the present paper, continuously, we improved our method and employed *trans*-β-Apo-8′-carotenal, an external standard reference which did not exist in the LB sample, instead of zeaxanthin dipalmitate to calculate the RCF. In order to estimate the feasibility of our hypothesis, we used β-carotene which existed in LB as another standard reference to calculate RCF as contrast. By using peak area combined with concentration, the values of RCF between *trans*-β-Apo-8′-carotenal/β-carotene and other four carotenoids including lutein, zeaxanthin, β-carotene, and zeaxanthin dipalmitate were calculated. Furthermore, by using our method, the carotenoid contents in 34 batches of LB samples collected from three regions of China were determined; the hierarchical cluster analysis (HCA), principal component analysis (PCA), and orthogonal partial least squares discriminant analysis (OPLS-DA) were employed to discriminate the different origins of LB. According to the Chinese Pharmacopeia 2020 (ChP2020), LB should be stored in cool and dry condition, protected from moisture and muggy conditions. To estimate the relationship between carotenoid contents and the storage conditions, the storage stabilities of carotenoids of LB under high humidity (RH 75 ± 5%) and high temperature (45 ± 5 °C) were also investigated.

## 2. Results and Discussion

### 2.1. Optimization of Chromatographic Conditions

To obtain a good separation among different nonpolar carotenoids in LB, besides the reverse phase chromatography column C30 which was applied, the chromatographic-grade dichloromethane was also used as one of the elution solvents. Meanwhile, due to the difficult preparation, only four carotenoids standard substances including lutein, zeaxanthin, β-carotene, and zeaxanthin dipalmitate were obtained. During the process of optimization of chromatographic conditions, we realized that the detection wavelength was crucial for developing a reliable QAMS method according to our previous studies, thus the prepared solutions including standard and sample solutions were all scanned over the entire UV range (200–800 nm). All the analytes showed proper absorptions at 450 nm. After repeated optimization, the chromatographic conditions were established as explained in “Section 3.2” below. Accordingly, the chromatograms of carotenoid standard substances and LB samples were displayed in Figure 1. Furthermore, the detailed information of the LB samples were presented in Table 1.

### 2.2. Calculation of Relative Correction Factor

To obtain the RCF of each analyte, an appropriate carotenoid was chosen as a standard reference, which was extremely vital. It was well known that most of the carotenoids are extremely unstable, and some of which could be isomerized when exposed to heat, light, even ambient temperature (25 ± 5 °C). Consequently, the measurement results were inaccurate due to the oxidation of the standard reference. After repeated experiments, *trans*-β-Apo-8′-carotenal was selected as a standard reference for its stable property, low price, and was easy to obtain. It was important to mention here that this compound was not present in LB, but its structure was similar to carotenoid. Meanwhile, to estimate the feasibility of our hypothesis, β-carotene, another carotenoid constituent which existed in LB was chosen as a contrast. According to “Equation (1)” in “Section 3.4”, the value of RCF for each analyte was calculated and presented in Table 2. Our data disclosed that by using *trans*-β-Apo-8′-carotenal as standard reference, the RCF of lutein, zeaxanthin, β-carotene, and zeaxanthin dipalmitate were 40.35 ± 0.11, 1.18 ± 0.001, 4.35 ± 0.02 and 2.60 ± 0.03, respectively; and the corresponding relative retention times (RRT) were 1.34, 1.41, 0.57, and 0.20, respectively. By employing β-carotene as the standard reference, the RCF of lutein, zeaxanthin, β-carotene, and zeaxanthin dipalmitate was 9.28 ± 0.06, 0.27 ± 0.00, 1.00 and 0.6 ± 0.01, respectively; the corresponding RRT was 2.37, 2.49, 1.00, and 0.36, respectively.

### 2.3. Validation of the Method

The QAMS methods were fully validated according to the guidelines of ChP2020, including linearity, limit of detection (LOD), limit of quantification (LOQ), accuracy, precision, stability, and robustness.

#### 2.3.1. Linearity, LOD and LOQ Tests

The calibration curves of the four standard substances were presented in Table 3. The concentration ranges of lutein, zeaxanthin, β-carotene, and zeaxanthin dipalmitate were 1.14–90.96 μg/mL, 0.082–20.42 μg/mL, 0.25–6.26 μg/mL, 1.01–91.15 μg/mL, respectively. All standard substances showed good linearity within the tested range of concentrations (*r* ≥ 0.9996). After a series of dilutions of each standard solution, the LOD and LOQ were determined based on signal-to-noise ratio (S/N) of about S/N = 3 and S/N = 10, respectively. The LOD and LOQ of the four carotenoids were ranged from 0.05 µg/mL to 0.57 µg/mL, and 0.082 µg/mL to 1.14 µg/mL, respectively.

#### 2.3.2. Accuracy Tests

The accuracy tests of the above four standard substances were all set at low (50% of the original amount), medium (100% of the original amount), and high (150% of the original amount) levels. Of note, almost all LB samples did not detect lutein, the added amount of lutein standards did not meet the above rules. The recovery of lutein, zeaxanthin, β-carotene, and zeaxanthin dipalmitate were 102.4%, 103.3%, 105.5%, and 96.0%, respectively, with corresponding relative standard deviation (RSD) were 7.8%, 6.8%, 6.1%, and 1.9%, respectively, as shown in Table 4. The results disclosed that except for zeaxanthin dipalmitate, the concentrations of lutein, zeaxanthin, and β-carotene were all extremely low in LB samples, which was closed to the LOQ, which thus led to low accuracy and high RSD in accuracy experiments.

#### 2.3.3. Precision Tests

Repeatability and reproducibility tests were employed to evaluate the precision of the method. The sample solution was injected 6 sequential times with RSD. Values of peak areas of zeaxanthin, β-carotene, and zeaxanthin dipalmitate were 0.57%, 1.25%, 0.21% respectively, which indicated a satisfactory chromatographic repeatability. The reproducibility was exerted by using six sample solutions which originated from same batch of LB sample; the RSD values of the concentrations of zeaxanthin, β-carotene, and zeaxanthin dipalmitate were 1.38%, 3.84% and 1.13%, respectively. The repeatability and reproducibility of the method were proved to be credible according to the results.

#### 2.3.4. Sample Solution Stability Tests

Concerning the stability of the sample solution, the peak areas of zeaxanthin, β-carotene, and zeaxanthin dipalmitate were measured at 0, 2, 4, 6, 8, 12, and 24 h, respectively, after preparation. The RSD values of zeaxanthin, β-carotene, and zeaxanthin dipalmitate were 2.72%, 4.74%, and 2.62%, respectively. β-carotene might be due to the low concentration in LB samples, thus leading to a high RSD value. The results indicated that the sample solution was stable within 24 h after preparation.

#### 2.3.5. Ruggedness Tests

To apply the established QAMS method in different laboratories, changes in instruments, columns, etc. were inevitable. In this study, the influences of different instruments, chromatographic columns, flow rates, and column temperatures on RCF were investigated. Our results, shown in Table 5, implied that different flow rates and column temperatures had no significant effects on the value of RCF. However, when using a C18 column, lutein and zeaxanthin could not be separated, implying that the use of chromatographic columns should be controlled in the separation of carotenoids in LB.

### 2.4. Quantitative Determination of Carotenoids in LB Samples

Four carotenoids including lutein, zeaxanthin, β-carotene, and zeaxanthin dipalmitate were measured in 34 batches of LB samples by using both external standard methods (ESM) and established QAMS methods (Table 6). The results revealed that all samples showed a high content of zeaxanthin dipalmitate, ranging from 0.81 mg/g to 4.05 mg/g, followed by zeaxanthin ranged from 0 to 28.17 µg/g. As the level of β-carotene was closed to LOQ, β-carotene was not detected in several batches, and none of LB samples contained detectable levels of lutein. Relatively, the highest contents of zeaxanthin dipalmitate, zeaxanthin, and β-carotene in LB samples were all seemly obtained from Qinghai. It was reported that LB was one of the substances with the highest contents of all-trans-zeaxanthin dipalmitate in nature, and LB from Qinghai could be used as one of its natural sources. Of note, the freshly picked LB had higher carotenoid contents; after being stored for a period, the color became darker, and the carotenoid contents decreased correspondingly.

To estimate the accuracy of our established QAMS methods, the standard method difference (SMD) between the ESM and QAMS method was determined, as shown in Table 7. The results disclosed that the SMD between ESM and QAMS ranged from 1.7% to 5.7% by using *trans*-β-Apo-8′-carotenal as a standard reference, and ranged from 0.1% to 9.5% by employing β-carotene as a standard reference. The difference between two standard references was due to the low concentration of β-carotene in LB. Our results disclosed that there was no significant difference among the results of the ESM and QAMS methods, which meant the established QAMS was feasible for the determination of carotenoid contents in LB by using *trans*-β-Apo-8′-carotenal as a standard reference.

According to the contents of zeaxanthin, β-carotene, and zeaxanthin dipalmitate in 34 batches of LB samples originating from three regions of China, the similarities of different LB samples were compared by using HCA, PCA, and OPLS-DA analysis. Concerning the HCA, as Figure 2 shows, all samples could be divided into two clusters: LB samples of Qinghai as one cluster, Ningxia and Gansu as another cluster. The results implied that the carotenoid contents of LB in Ningxia were consistent with the LB in Gansu, but both of them were significantly different from LB collected from Qinghai. Looking at the geographical position of those three provinces in China, Ningxia is adjacent to Gansu, whereas Gansu is adjacent to Qinghai. Specifically, the location of LB collected from Guansu province was Baiyin city whose latitude and longitude was 104.682515°, 36.577096°, which was adjacent to Zhongning city of Ningxia with latitude and longitude was 105.691537°, 37.497421°, while LB obtained from Qinghai province was Delingha city whose latitude and longitude was 96.719684°, 37.356337°, which was in the center of Qinghai and far from Baiyin and Zhongning. Therefore, the carotenoid contents of LB were mainly affected by the geographical position, especially the location of latitude. To obtain the overall characteristics of carotenoids in LB samples from different regions, PCA and OPLS-DA tests were performed, and the results were shown in Figure 3. Similarly, the LB samples could be clearly differentiated into two groups in both the PCA and OPLS-DA models; one group contained LB samples were mainly from Ningxia and Gansu provinces, and the other group was mainly from Qinghai, which was consistent with the results of HCA. Therefore, no matter what statistical methods were applied, the LB samples of Gansu were similar to that of Ningxia, while both of them were different to the LB samples of Qinghai. Thus, we assumed that the carotenoids might be used as markers for the regional characterization of LB.

### 2.5. Storage Condition and Stable Evaluation

According to ChP2020, LB was characterized by large grains, red color, thick flesh, few seeds, soft texture, and sweet taste, and color was an important index for its character evaluation. It was well known that color played an important role in consumer choices, which was directly related to quality and authenticity, and might also be related to the presence of specific chemical components (pigments) [19]. As shown in Figure 4, under high humidity (RH 75 ± 5%) and high temperature (45 ± 5 °C) conditions, the content of zeaxanthin dipalmitate was significantly decreased within the storage time as compared with day 0, while under the ambient temperature (25 ± 5°C), this change was slight lower than in high temperature and high humidity. From day 0 to day 28, with the prolongation of storage time, zeaxanthin dipalmitate in LB could not be detected under high humidity and high temperature conditions. Interestingly, we found that the content of zeaxanthin dipalmitate was significantly decreased as the color of LB changed from bright red to black-brown. Concerning the decrease rate presented in Figure 4D, under high humidity (RH 75 ± 5%) and high temperature (45 ± 5 °C) conditions, the decrease rates of zeaxanthin dipalmitate were similar, which were all higher than under ambient temperature (25 ± 5 °C). However, after 28 days of storage, all the decrease rates of zeaxanthin dipalmitate ultimately reached about 80%, regardless of the storage temperature and humidity. The results of this study are of great significance for quality evaluation of storage and maintenance of LB.

## 3. Materials and Methods

### 3.1. Reagents and Chemicals

Lutein (AF8033002, purity > 95%), zeaxanthin (AF8062716, purity 98%), and *trans*-β-Apo-8′-carotenal (AF9100402, purity 98%) were purchased from Chengdu Alfa Biotechnology Co., Ltd., Chengdu, China. β-Carotene (10445-201802, the purity of 99.8%) was obtained from National for Food and Drug control, Beijing, China. Zeaxanthin dipalmitate (04160, purity > 95%) was supplied by Extrasynthese Chemical S.A.S., Lyon, France. The structures of the 5 carotenoids were shown in Figure 1. Methanol, acetonitrile, and dichloromethane were all chromatographic grade obtained from Fisher Scientific, Nepean, Canada. Acetone, *n*-hexane, toluene, 2,6-di-*tert*-butyl-4-methylphenol (BHT), and anhydrous ethanol were all at analytical grade and purchased from Tianjin Damao Chemical Reagent Factory, Tianjin, China. Double-distilled water was used. A total of 34 batches of LB were collected from 3 different regions of China, including Qinghai (96.719684°, 37.356337°), Gansu (104.682515°, 36.577096°), and Ningxia (105.691537°, 37.497421°). LB samples were identified by Prof. Ling Dong (Department of Pharmacognosy, Ningxia Medical University), with the corresponding voucher specimens (numbered NYYP20190701-34) were preserved in the herbarium of pharmaceutical analysis. The detailed information of the samples was presented in Table 1.

### 3.2. Instruments and Chromatographic Conditions

HPLC-DAD analyses were performed on an Agilent 1260 Infinity HPLC (Agilent Technologies, Stockport, UK) system (equipped with 1260 Quat Pump VL, 1260 Vialsampler and 1260 DAD WR) by using a column of C30 (YMC, 4.6 × 250 mm, 5 μm). A gradient of mobile phase was used for efficient separation, mobile phases A (dichloromethane) and B (methane: acetonitrile: water, 81:14:5, *v*/*v*/*v*), with the elution program was as follows: 0–20 min, 30% A; 20–48 min, 50% A; 48–50 min, 70% A; 50–55 min, 70% A. The flow rate was 1 mL/min with the column temperature maintained at 22 °C. Detection wavelength was 450 nm, and the sample injection volume was 20 µL.

### 3.3. Preparation of Standard and Sample Solutions

Standard solutions were freshly prepared by dissolving approximately 10 mg of each standard substance, including *trans*-β-Apo-8′-carotenal, lutein, zeaxanthin, β-carotene, and zeaxanthin dipalmitate, into a 10 mL brown volumetric flask, respectively, with dichloromethane as the solvent, and then diluted with anhydrous ethanol which contained 0.1% BHT (1000-fold dilution) to obtain different solutions of each standard for the determination of RCF using HPLC method. To prevent the volatilization of dichloromethane, the flasks were sealed with parafilm and stored at 4 °C.

LB samples were first dried under 60 °C conditions for 12 h, powdered, and sieved through an 80-mesh sieve after cooling to ambient temperature (25 ± 5 °C). After mixing with a blender, approximately 1 g of LB powders was accurately weighted, added into a 20 mL brown volumetric flask with a mixed solvent including hexane, ethanol, acetone, and toluene (10:6:7:7, *v*/*v*/*v*/*v*), extracted using an Elmasonic P 120H ultrasonic bath (Elma, Germany) for 30 min at ambient temperature (25 ± 5 °C), then filtered and diluted with anhydrous ethanol which contained 0.1% BHT (50-fold dilution) and *trans*-β-Apo-8′-carotenal standard reference (approximately 1.0 µg/mL) for HPLC analysis.

### 3.4. Calculation of Relative Correction Factor, Relative Retention Time and Quantification of Carotenoids in Different LB Samples

In the present study, by applying *trans*-β-Apo-8′-carotene and β-carotene as the standard references, the value of RCF was calculated by the ratio of the concentration and HPLC peak area between *trans*-β-Apo-8′-carotenal/β-carotene and the other analyte, as Equation (1) showed. Of note, the RCF for each carotenoid was calculated by using three concentrations respectively, which all fell within the ranges of the calibration curve. These three concentrations were selected based on the highest, middle, and lowest contents of each carotenoid in 34 batches of LB. The relative retention time (RRT) was calculated followed by Equation (2). The quantification of bioactive carotenoids from LB could be carried out according to the following Equations (3) and (4).
(1)$$f_{s/x}=\frac{f_{s}}{f_{x}}=\frac{C_{s\;}}{C_{x}}\times \frac{A_{x}}{A_{s}}$$
(2)$$RRT_{s/x}=\frac{t_{s}}{t_{x}}$$
(3)$$C\prime_{x}=\frac{A\prime_{x}\times C_{s}\times f_{s/x}}{A_{s}}$$
(4)$$W_{x}=\frac{C\prime_{x}\times V}{m}$$
where *A_s_* and *C_s_* were the peak areas and concentrations of the standard reference, respectively. *A_x_* and *C_x_* were the peak areas and concentrations of the analyte to be measured in the standard solution, respectively. *t_x_* and *t_s_* were the retention times of the analyte and standard reference to be measured in the standard solution, respectively. *A*′*_x_* and *C*′*_x_* were the peak areas and concentrations of the analyte in sample solution, respectively. *V* was the volume of the sample solution; *m* was the mass of the sample; *W_x_* was the mass concentration of analyte in the sample. For the comparison of this new QAMS method with ESM, the SMD was calculated according to the following Equation (5).
(5)$$SMD=\left|\frac{W_{ESM}-W_{QAMS}}{W_{ESM}}\right|$$
where *W_ESM_* and *W_QAMS_* were the mass concentration of analyte in LB sample measured by using ESM and QAMS methods, respectively.

For the method validation, ChP2020 guidelines on the validation of analytical methods were followed, the main items were containing linearity, LOD, LOQ, accuracy, precision, stability and robustness.

Based on the above established QAMS method, and compared with ESM method, four carotenoids, including lutein, zeaxanthin, β-carotene, and zeaxanthin dipalmitate, were measured in 34 batches of LB samples from 3 regions of China. To estimate the feasible of our method, the SMD between two methods were also evaluated.

Meanwhile, in order to compare the similarities of LB samples collected from different regions of China based on the carotenoid contents, different statistical methods including HCA, PCA, and OPLS-DA were employed.

### 3.5. Stability Evaluation of LB

In daily life, it was found that the color of LB always became dark during the storage time, especially under high humidity (RH > 60%) and high temperature (>30 °C), implied the changes in the pigment composition of LB. Given zeaxanthin dipalmitate was the main pigment in LB, the stability of LB under ambient temperature, high temperature and high humidity conditions were estimated. Based on the “Technical Guidelines for the Stability Research of Traditional Chinese Medicine and Natural Medicines” of ChP2020, the 34 batches of LB were stored in different stability conditions including high humidity (RH 75 ± 5%) with ambient temperature (25 ± 5 °C), high temperature (45 ± 5 °C) with normal humidity (50 ± 5%), and ambient temperature (25 ± 5 °C) with normal humidity (50 ± 5%), respectively. Sampling and testing were performed at 0, 7 d, 14 d, 21 d and 28 d, respectively. The sample solution was prepared according to the above process described in “Section 3.3” above, and the content of zeaxanthin dipalmitate was calculated using the established QAMS method which employed trans-β-Apo-8′-carotene as standard reference.

### 3.6. Data Analysis

By employing SPSS 26.0 and SMICA 14.1 software, LB samples from different geographical origins of China were analyzed.

## 4. Conclusions

In the present study, a new QAMS method was established by using *trans*-β-apo-8′-carotenal, a substance that did not exist in the LB sample, as a standard reference to determine the four carotenoids in LB simultaneously. The small SMD between ESM and QAMS methods implied the feasibility of our method. By establishing the RCF between the standard reference and carotenoids, the quantities of the carotenoids can be directly calculated in practical applications, thus the QAMS method provided a more convenient, faster, cheaper, and simpler way for evaluating the quality of LB. Based on the carotenoid content, 34 batches of LB samples could be clearly divided into two groups by HCA, PCA, and OPLS-DA analysis (Group 1: Qinghai; Group 2: Ningxia and Gansu), which was directly related to the geographic locations of the different LB samples. The storage stability test of LB implied that zeaxanthin dipalmitate content decreased significantly as the color of LB changed from bright red to black-brown under high humidity and high temperature conditions, even at the ambient temperature.

## Figures and Tables

**Figure 1 molecules-26-05374-f001:**
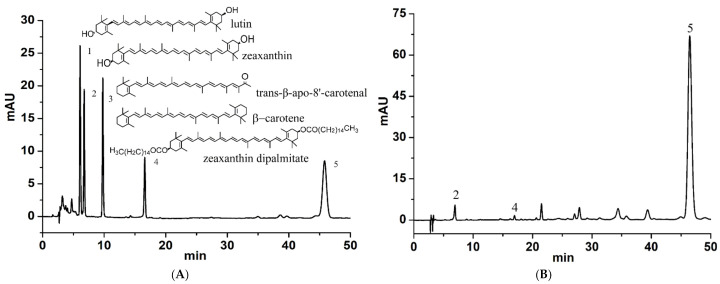
HPLC chromatograms and structures of carotenoids. (**A**) chromatogram of mixed standard solutions; 1—lutein, 2—zeaxanthin, 3—*trans*-β-Apo-8′-carotenal, 4—β-carotene, 5—zeaxanthin dipalmitate. (**B**) chromatogram of LB sample solution; 2—zeaxanthin, 4—β-carotene, 5—zeaxanthin dipalmitate.

**Figure 2 molecules-26-05374-f002:**
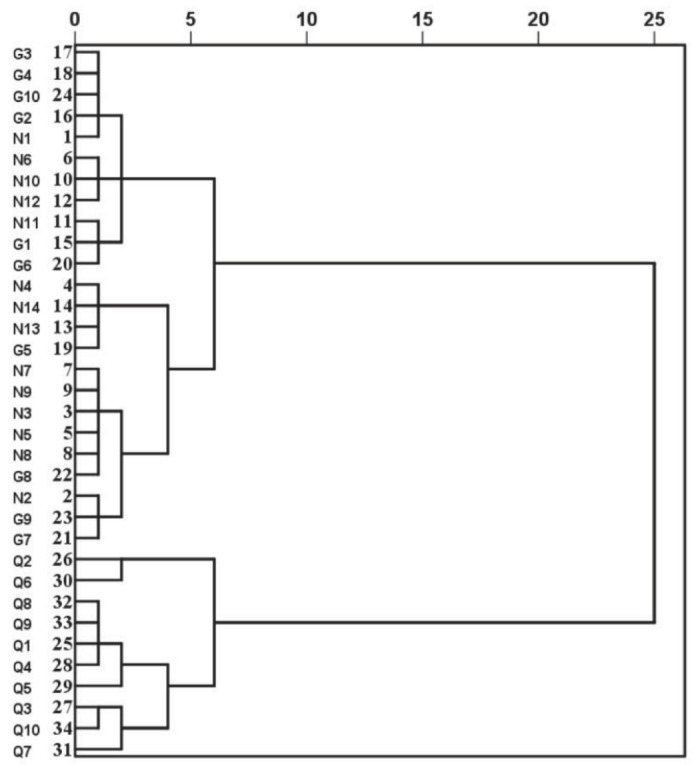
HCA dendrogram of 34 batches of LB. The abscissa indicated the squared euclidean distance, whereas the ordinate expressed the sample numbers.

**Figure 3 molecules-26-05374-f003:**
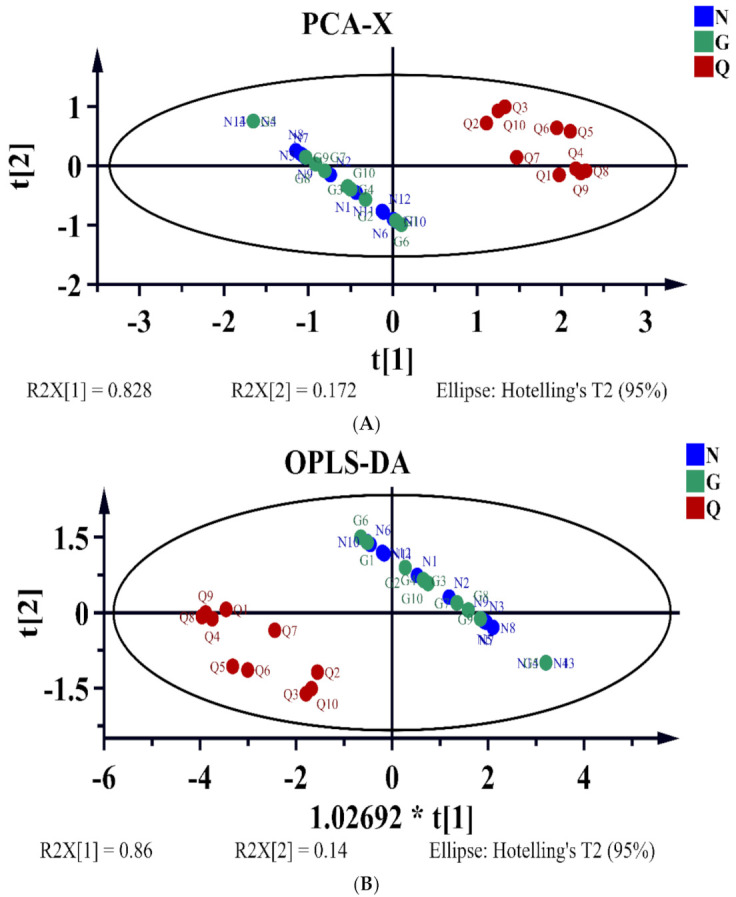
PCA and OPLS-DA plots of 34 batches of LB. (**A**) the PCA-X plot of 34 batches of LB. When zeaxanthin dipalmitate was used as Y variable, β-carotene, and zeaxanthin as X variable, the first two principal components explained 100% of variance (PC1 represented 82.8% and PC2 represented 17.2%) based on the three variables of carotenoids. (**B**) the OPLS-DA plot of 34 batches of LB. R^2^Y and R^2^Q were 76% and 72.8%, respectively, implying a great of reliability and predictability of the model.

**Figure 4 molecules-26-05374-f004:**
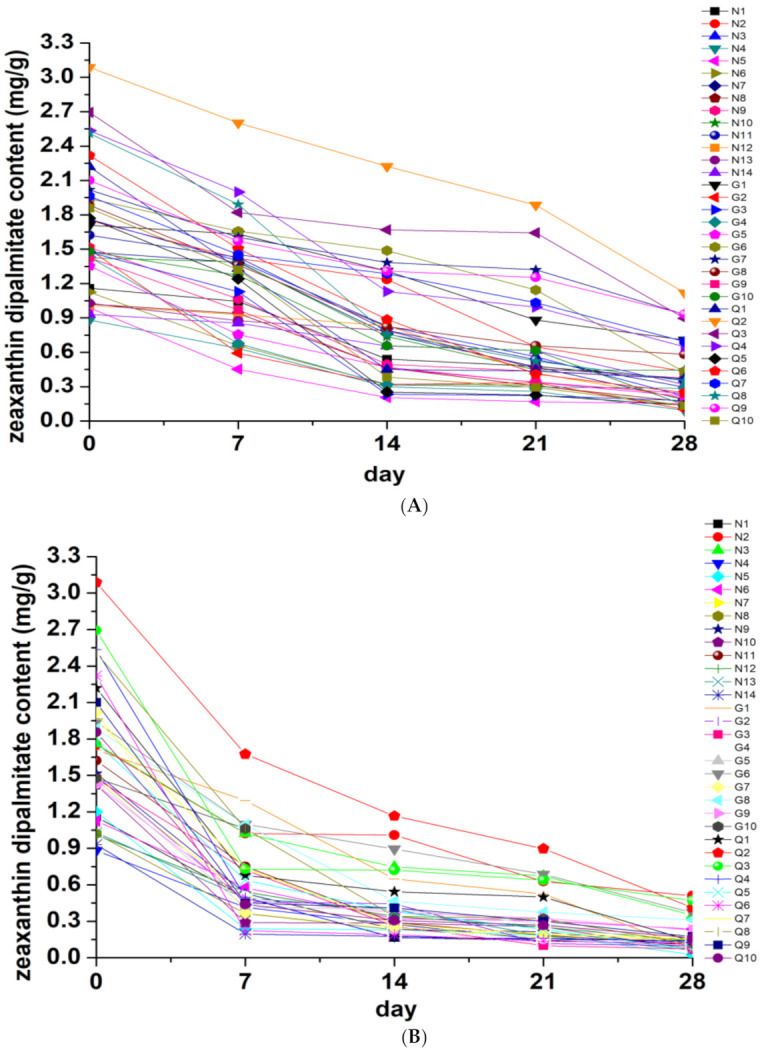
Determination results and decrease rates of zeaxanthin dipalmitate (mg/g) in 34 batches of LB under different storage conditions from day 0 to day 28. (**A**) content of zeaxanthin dipalmitate (mg/g) in LB under ambient temperature (25 ± 5 °C); (**B**) content of zeaxanthin dipalmitate (mg/g) in LB under high temperature (45 ± 5 °C); (**C**) content of zeaxanthin dipalmitate (mg/g) in LB under high humidity (RH 75 ± 5%); (**D**) decrease rates of zeaxanthin dipalmitate in LB under ambient temperature, high temperature and high humidity, respectively.

**Table 1 molecules-26-05374-t001:** The information of 34 batches of LB.

NO	Origin	Batch	NO	Origin	Batch
N1	Ningxia	20190917	G4	Gansu	20190974
N2	Ningxia	20190918	G5	Gansu	20190970
N3	Ningxia	20180920	G6	Gansu	20190924
N4	Ningxia	20190921	G7	Gansu	20190910
N5	Ningxia	20190922	G8	Gansu	20190115
N6	Ningxia	20190935	G9	Gansu	20190956
N7	Ningxia	20190904	G10	Gansu	20190957
N8	Ningxia	20190908	Q1	Qinghai	20191109
N9	Ningxia	20190911	Q2	Qinghai	20190931
N10	Ningxia	20190925	Q3	Qinghai	20191041
N11	Ningxia	20190927	Q4	Qinghai	20190113
N12	Ningxia	20180903	Q5	Qinghai	20191110
N13	Ningxia	20170708	Q6	Qinghai	20191044
N14	Ningxia	20190950	Q7	Qinghai	20191111
G1	Gansu	20190936	Q8	Qinghai	20191045
G2	Gansu	20190930	Q9	Qinghai	20191055
G3	Gansu	20190952	Q10	Qinghai	20191035

**Table 2 molecules-26-05374-t002:** RCF values calculated by using *trans*-β-Apo-8′-carotenal and β-carotenal as standard references.

Standard Substance	*trans*-β-Apo-8′-Carotenal as Reference		β-Carotene as Reference	
	RCF	RRT	RCF	RRT
Lutein	40.35 ± 0.11	1.34	9.28 ± 0.06	2.37
Zeaxanthin	1.18 ± 0.01	1.41	0.27 ± 0.00	2.49
β-Carotene	4.35 ± 0.02	0.57	1.00	1.00
Zeaxanthin dipalmitate	2.60 ±0.03	0.20	0.6 ± 0.01	0.36

**Table 3 molecules-26-05374-t003:** Calibration curve, LOQ and LOD of standard substances.

Standard Substance	Calibration Curve	*R* ^2^	Test Range μg/mL	LOQ μg/mL	LOD μg/mL
Lutein	y = 6.2046x − 5.3418	0.9996	1.14–90.96	1.14	0.57
Zeaxanthin	y = 139.72x + 34.43	0.9995	0.082–20.42	0.082	0.05
β-Carotene	y = 49.31x + 6.046	0.9995	0.25–6.26	0.25	0.13
Zeaxanthin dipalmitate	y = 98.505x − 40.763	0.9993	1.01–91.15	1.01	0.44

**Table 4 molecules-26-05374-t004:** The RSD of precision, repeatability, and accuracy of carotenoids in LB samples (%).

Compound	Repeatability	Reproducibility	Stability	Recovery/RSD
Lutein	0	0	0	102.4/7.8
Zeaxanthin	0.57	1.38	2.72	103.3/6.8
β-Carotene	1.25	3.84	4.74	105.5/6.1
Zeaxanthin dipalmitate	0.21	1.13	2.62	96.0/1.9

**Table 5 molecules-26-05374-t005:** The influence of different factors on the value of RCF of each component.

Factor	Level/Brand	Lutein	Zeaxanthin	β-Carotene	Zeaxanthin Dipalmitate
Different flow rates	0.8 min/mL	40.49	1.21	4.94	2.61
	0.9 min/mL	40.53	1.21	4.64	2.64
	1.1 min/mL	40.52	1.21	4.82	2.65
Different temperatures	23 °C	41.04	1.24	5.30	2.83
	25 °C	40.95	1.21	5.36	2.72
	30 °C	43.10	1.19	5.23	2.65
Different columns	YMC C30	40.35	1.18	4.35	2.60
	UG17546250W C30	40.11	1.15	4.26	2.58
Different instruments	Hitachi2000	38.00	1.23	5.83	3.03
	Waters2998	38.47	1.09	4.01	2.86
	Shimadzu 2030C	42.73	1.23	5.46	2.93

**Table 6 molecules-26-05374-t006:** The carotenoid contents in LB samples by using ESM and QAMS methods.

Sample	Zeaxanthin Dipalmitate (mg/g)			Zeaxanthin (μg/g)			β-Carotene (μg/g)		
	ESM	QAMS1	QAMS2	ESM	QAMS1	QAMS2	ESM	QAMS1	QAMS2
N1	1.05	1.11	1.09	14.32	14.25	14.54	––	––	––
N2	1.71	1.74	1.71	10.77	10.94	11.16	––	––	––
N3	1.58	1.61	1.58	6.87	6.84	6.98	––	––	––
N4	0.81	0.86	0.84	––	––	––	––	––	––
N5	1.20	1.22	1.20	6.82	6.79	6.93	––	––	––
N6	1.18	1.20	1.18	19.50	18.97	19.36	––	––	––
N7	1.46	1.48	1.45	6.60	6.57	6.70	––	––	––
N8	1.16	1.18	1.16	5.88	5.72	5.84	––	––	––
N9	1.46	1.49	1.46	7.24	7.21	7.36	––	––	––
N10	1.29	1.31	1.28	19.94	19.39	19.79	––	––	––
N11	1.75	1.78	1.75	17.97	17.47	17.83	––	––	––
N12	0.93	0.98	0.96	18.18	17.68	18.04	––	––	––
N13	1.22	1.24	1.22	––	––	––	––	––	––
N14	0.87	0.91	0.89	––	––	––	––	––	––
G1	1.75	1.78	1.75	19.86	19.32	19.71	––	––	––
G2	1.40	1.48	1.45	15.66	15.23	15.54	––	––	––
G3	1.29	1.31	1.28	13.54	13.17	13.44	––	––	––
G4	1.28	1.30	1.27	13.62	13.24	13.51	––	––	––
G5	1.37	1.39	1.36	––	––	––	––	––	––
G6	2.07	2.11	2.07	20.62	20.05	20.46	––	––	––
G7	2.35	2.39	2.34	9.92	9.87	10.07	––	––	––
G8	1.19	1.25	1.23	8.63	8.59	8.77	––	––	––
G9	1.97	2.01	1.97	7.30	7.26	7.41	––	––	––
G10	1.32	1.35	1.32	13.11	13.05	13.32	––	––	––
Q1	2.94	3.03	2.97	26.76	26.64	27.18	6.08	5.93	6.59
Q2	4.05	4.17	4.09	16.56	16.48	16.82	6.11	5.96	6.62
Q3	2.50	2.58	2.53	16.19	16.11	16.44	7.21	7.03	7.81
Q4	2.57	2.65	2.60	27.30	26.55	27.09	6.78	6.61	7.34
Q5	2.81	2.89	2.83	23.16	22.52	22.98	8.04	7.84	8.71
Q6	3.84	3.95	3.87	21.86	21.26	21.69	7.81	7.62	8.47
Q7	2.54	2.62	2.57	22.01	21.40	21.84	5.62	5.48	6.09
Q8	3.04	3.13	3.07	28.17	27.40	27.96	6.97	6.80	7.56
Q9	3.08	3.17	3.11	28.06	27.29	27.85	6.73	6.56	7.29
Q10	2.74	2.82	2.76	16.11	15.67	15.99	6.89	6.72	7.47

QAMS1: *trans*-β-Apo-8′-carotenal as standard reference to obtain RCF; QAMS2: β-carotene as standard reference to obtain RCF.

**Table 7 molecules-26-05374-t007:** The SMD between different QAMS methods with ESM (%).

Sample	Zeaxanthin Dipalmitate		Zeaxanthin		β-Carotene	
	SMD1	SMD2	SMD1	SMD2	SMD1	SMD2
N1	5.7	3.6	0.5	1.5	––	––
N2	1.8	0.2	1.6	3.7	––	––
N3	1.9	0.1	0.4	1.6	––	––
N4	6.2	4.1	––	––	––	––
N5	1.7	0.3	0.4	1.6	––	––
N6	1.7	0.3	2.7	0.7	––	––
N7	1.4	0.6	0.5	1.6	––	––
N8	1.7	0.3	2.7	0.7	––	––
N9	2.1	0.1	0.4	1.6	––	––
N10	1.6	0.4	2.8	0.8	––	––
N11	1.7	0.3	2.8	0.8	––	––
N12	5.4	3.3	2.8	0.8	––	––
N13	1.6	0.4	––	––	––	––
N14	4.6	2.5	––	––	––	––
G1	1.7	0.3	2.7	0.7	––	––
G2	5.7	3.6	2.7	0.8	––	––
G3	1.6	0.4	2.7	0.7	––	––
G4	1.6	0.4	2.8	0.8	––	––
G5	1.5	0.5	––	––	––	––
G6	1.9	0.1	2.8	0.8	––	––
G7	1.7	0.3	0.5	1.5	––	––
G8	5.0	3.0	0.5	1.6	––	––
G9	2.0	0.0	0.5	1.5	––	––
G10	2.3	0.3	0.5	1.6	––	––
Q1	3.1	1.0	0.4	1.6	2.5	8.4
Q2	3.0	0.9	0.5	1.5	2.5	8.4
Q3	3.2	1.2	0.5	1.5	2.5	8.3
Q4	3.1	1.1	2.7	0.8	2.5	8.3
Q5	2.8	0.8	2.8	0.8	2.5	8.3
Q6	2.9	0.8	2.7	0.8	2.4	8.4
Q7	3.1	1.1	2.8	0.8	2.5	8.3
Q8	3.0	0.9	2.7	0.7	2.4	8.4
Q9	2.9	0.9	2.7	0.8	2.5	8.3
Q10	2.9	0.9	2.7	0.7	2.5	8.4

SMD1: SMD between ESM and QAMS 1 (*trans*-β-Apo-8′-carotenal as standard reference to obtain RCF); SMD2: SMD between ESM and QAMS 2 (β-carotene as standard reference to obtain RCF).

## Data Availability

The data used to support the findings of this study are available from the corresponding author upon request.

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
