# Peer review of "Carotenoid Contents of Lycium barbarum: A Novel QAMS Analyses, Geographical Origins Discriminant Evaluation, and Storage Stability Assessment"

_molecules, 2021, doi:10.3390/molecules26175374_

Round 1
Reviewer 1 Report
In the paper „Carotenoid content of Lycium barbarum: a novel QAMS analyses, geographical origins discriminant evaluation, and storage stability assessment” was established a new QAMS method which use trans-β-apo-8'-carotenal as standard reference for the determination four carotenoids from Lycium barbarum and then, the method was validated. Also, depending on the dipalmitate content of zeaxanthin, the 34 batches of Lycium barbarum samples used could be divided into two groups, which are consistent with the geographical locations where the samples come from. The content of zeaxanthin dipalmitate (mg/g) was determined and its decrease in storage was observed, a decrease that depends on the storage conditions.
Strengths: validation of a method for carotenoid determination in Lycium barbarum, variation of dipalmitate zeaxanthin content depending on storage conditions
Suggestion: examples of other determinations of Lycium barbarum carotenoids performed by other methods
Author Response
Response: Thank you for your warm suggestion. In our initial manuscript, to estimate the accuracy of the established QAMS method, the external standard method for the determinations of Lycium barbarum carotenoids was used as contrast, and the low standard method difference between two methods proved the feasibility of our method.
All the changes we made in the manuscript were highlighted in red colored text.

Reviewer 2 Report
The manuscript “Carotenoid content of Lycium barbarum: a novel QAMS analyses…” shows an interesting research work about the quantitation of carotenoids in natural samples from different origins. Before its acceptation for publication in “Molecules” the following items should be considered:
1.- English must be revised throughout the manuscript. There are numerous mistakes by changing adjectives by substantives or vice versa, or misuses of verb tenses. As a small sample, in line 19 “by employed” should be changed to “by employing”; line 20 “standard references” -> “standard reference”. The following sentence in line 20 has no sense; line 25 “samples were collected” -> “samples collected”; line 28 “the decrease rates were all reached” -> “all the decrease rates reached”; line 59 “efficacy” -> “efficient”; line 60 “was need” -> “was needed”; line 64 “official” -> “officially”. The rest of the manuscript is not very different to this, thus making difficult the understanding of several sentences or sections.
2.- Despite described at the end of the manuscript in a detailed section, all the abbreviations should be defined when used for the first time (e.g. TCM in line 38).
3.- Authors mention the QAMS principle and define the different approaches. After a careful reading of the manuscript, the final chosen option is a little bit unclear. Only one concentration has been considered for the calculation of relative factors? What is the concentration selected? According to the results of table 3, it is clear that the slopes of the different considered compounds are quite different, so the RCF must vary with concentration. Have authors considered the possibility of matching RCF value as a function of the concentration of the reference compound instead of using an only value?
4.- In most cases “content” should be replaced by “concentration”
5.- If chromatographic separation has been carried out with a C30 column, all the references (including ruggedness study) to C18 columns should be deleted, since they are not relevant. In addition, 50 min for the separation of only 5 compounds is a very long time. An appropriate selection of solvents/temperature could greatly improve the analysis time.
6.- In line 117 “sample solution (sample N5 from Ningxia)” is mentioned, but it is the same than the rest of “sample solution” which appear in different parts of the manuscript. This must be clearly specified in all the cases, if it corresponds to a natural sample or a synthetic one.
7.- In line 147 “a wide range of concentrations” in most cases do not reach 2 magnitude orders.
8.- In table 3, the intercept seems to be very high, and can be very relevant at low concentrations. In addition, the lower value of the range of lutein (5.66) should be 1.14 (LOQ).
9.- In line 158, ranges for recoveries are reported, but in table 4 only one value is given, and do not correspond to the previous ones. Please, revise. Values reported in line 172 do not appear in the table 4.
10.- In line 180, “low RSD value” should be “high RSD value”
11.- What mean exactly the values reported in table 5? In addition, it is unclear the rest of conditions used for each experiment, e.g. when testing different instruments, what column/flow/temperature was chosen?
12.- Lines 201-203. The colour changed due to the storing period, but the coordinates (e.g. RGB) are not reported. Maybe it could be also a variable to be considered in statistical tests.
13.- In line 213, no significant differences were observed, according to the authors, from SMD. What is the critical value? The underlined values in table 7 simply correspond to the highest value of the column?
14.- In line 231, “Delinha” is really “Delingha”. Please, change also “was latitude” to “whose latitude”.
15.- Line 298: please, replace “methane” to “methanol”
16.- Lines 305-306: Both dichloromethane and ethanol are very volatile solvents. Once prepared the standard solutions, the volume/mass was checked regularly along time to control unnoticed evaporation of solvent, thus increasing the concentration of solution?
17.- In section 3.4 the relative retention time is not described, as it could be expected from the title.
18.- Line 355: High humidity was applied at high temperature? Ambient temperature? Both?
Author Response
Review #2:
The manuscript “Carotenoid content of Lycium barbarum: a novel QAMS analyses…” shows an interesting research work about the quantitation of carotenoids in natural samples from different origins. Before its acceptation for publication in “Molecules” the following items should be considered:
1.- English must be revised throughout the manuscript. There are numerous mistakes by changing adjectives by substantives or vice versa, or misuses of verb tenses. As a small sample, in line 19 “by employed” should be changed to “by employing”; line 20 “standard references” -> “standard reference”. The following sentence in line 20 has no sense; line 25 “samples were collected” -> “samples collected”; line 28 “the decrease rates were all reached” -> “all the decrease rates reached”; line 59 “efficacy” -> “efficient”; line 60 “was need” -> “was needed”; line 64 “official” -> “officially”. The rest of the manuscript is not very different to this, thus making difficult the understanding of several sentences or sections.
Response: Thank you for your warm suggestion. We apologize for our careless mistakes. We invited a native English speaking colleague to help us checking all of the grammar, spelling, punctuation and style of our initial manuscript. Accordingly, we have made an extensive and careful revision in our revised manuscript. All the changes we made in the manuscript were highlighted in red color.
2.- Despite described at the end of the manuscript in a detailed section, all the abbreviations should be defined when used for the first time (e.g. TCM in line 38).
Response: Thank you for your careful suggestion. We complemented the definition of abbreviation including TCM, ESM and SMD when used for the first time.
3.- Authors mention the QAMS principle and define the different approaches. After a careful reading of the manuscript, the final chosen option is a little bit unclear. Only one concentration has been considered for the calculation of relative factors? What is the concentration selected? According to the results of table 3, it is clear that the slopes of the different considered compounds are quite different, so the RCF must vary with concentration. Have authors considered the possibility of matching RCF value as a function of the concentration of the reference compound instead of using an only value?
Response: Thank you for your professional and insightful comments. In our initial manuscript, the RCF of each carotenoid was calculated by using three concentrations respectively, which all fell within the ranges of the calibration curves. These three concentrations were selected based on the highest, middle and lowest contents of each carotenoid in 34 batches of LB. Thus, the concentrations of zeaxanthin selected for calculation of RCF were 0.082 μg/mL, 0.408 μg/mL and 1.021 μg/mL, respectively; β-carotene were 0.25 μg/mL, 0.51 μg/mL and 1.00μg/mL respectively; and zeaxanthin dipalmitate were 10.09 μg/mL, 15.58 μg/mL and 91.15 μg/mL, respectively. Accordingly, we complemented the description in our revised manuscript as follows: ……Of note, the RCF for each carotenoid was calculated by using three concentrations respectively, which all fell within the ranges of the calibration curve. These three concentrations were selected based on the highest, middle and lowest contents of each carotenoid in 34 batches of LB……
As you said, “the slopes of the different considered compounds are quite different, so the RCF must vary with concentration”. When we prepared our initial manuscript, the slope of the standard reference was also considered for the calculation of RCF, as the blow table 1 shown. We found that the RCF value varied between different calculation approaches, which directly influence the accuracy of the determination. In order to observe the standard method difference between different QAMS methods with external standard method, the carotenoid contents in LB samples by using different QAMS methods were also estimated, the results were presented in the follow table 2 and table 3. It was found that when using slopes to calculate the RCF, large differences were found between QAMS and ESM. Maybe a wide concentration range used in calibration curve is the responsibility.
According to our previous efforts, we had established several ways to calculate RCF, one was determined by the ratio of the concentrations and HPLC peak areas between the standard reference and analyte, the second was the proportion of the slopes established by chromatographic/spectra calibration curve, the third was the proportion of the absorption coefficients. Your insightful suggestions will provide new ideas and concepts for our future studies. Thank you again.
Table 1 RCF values calculated by using trans-β-Apo-8’-carotenal and β-carotenal as standard reference
|
compounds |
trans-β-Apo-8’-carotenal as reference |
β-carotenal as references |
||
|
fs/x (peak area) |
fs/x (slope) |
fs/x (peak area) |
fs/x (slope) |
|
|
lutein |
40.35±0.11 |
38.80 |
9.37±0.06 |
7.95 |
|
zeaxanthin |
1.18±0.01 |
1.72 |
0.27±0.00 |
0.35 |
|
β-carotene |
4.35±0.02 |
4.88 |
1.00 |
1.00 |
|
zeaxanthin dipalmitate |
2.60±0.03 |
2.44 |
0.6±0.01 |
0.50 |
Table 2 The content of carotenoids in LB samples by using ESM and QAMS methods
|
sample |
zeaxanthin dipalmitate(mg/g) |
zeaxanthin(μg/g) |
β-carotene(μg/g) |
||||||||||||||
|
ESM |
QAMS1 |
QAMS2 |
QAMS3 |
QAMS4 |
ESM |
QAMS1 |
QAMS2 |
QAMS3 |
QAMS4 |
ESM |
QAMS1 |
QAMS2 |
QAMS3 |
QAMS4 |
|||
|
N1 |
1.05 |
1.11 |
1.04 |
1.09 |
0.91 |
14.32 |
14.25 |
20.77 |
14.54 |
18.85 |
–– |
–– |
–– |
–– |
–– |
||
|
N2 |
1.71 |
1.74 |
1.63 |
1.71 |
1.42 |
10.77 |
10.94 |
15.95 |
11.16 |
14.47 |
–– |
–– |
–– |
–– |
–– |
||
|
N3 |
1.58 |
1.61 |
1.51 |
1.58 |
1.32 |
6.87 |
6.84 |
9.97 |
6.98 |
9.05 |
–– |
–– |
–– |
–– |
–– |
||
|
N4 |
0.81 |
0.86 |
0.81 |
0.84 |
0.70 |
–– |
–– |
–– |
–– |
–– |
–– |
–– |
–– |
–– |
–– |
||
|
N5 |
1.20 |
1.22 |
1.14 |
1.20 |
1.00 |
6.82 |
6.79 |
9.90 |
6.93 |
8.98 |
–– |
–– |
–– |
–– |
–– |
||
|
N6 |
1.18 |
1.20 |
1.13 |
1.18 |
0.98 |
19.50 |
18.97 |
27.65 |
19.36 |
25.09 |
–– |
–– |
–– |
–– |
–– |
||
|
N7 |
1.46 |
1.48 |
1.39 |
1.45 |
1.21 |
6.60 |
6.57 |
9.58 |
6.70 |
8.69 |
–– |
–– |
–– |
–– |
–– |
||
|
N8 |
1.16 |
1.18 |
1.11 |
1.16 |
0.96 |
5.88 |
5.72 |
8.34 |
5.84 |
7.57 |
–– |
–– |
–– |
–– |
–– |
||
|
N9 |
1.46 |
1.49 |
1.40 |
1.46 |
1.22 |
7.24 |
7.21 |
10.51 |
7.36 |
9.54 |
–– |
–– |
–– |
–– |
–– |
||
|
N10 |
1.29 |
1.31 |
1.23 |
1.28 |
1.07 |
19.94 |
19.39 |
28.26 |
19.79 |
25.65 |
–– |
–– |
–– |
–– |
–– |
||
|
N11 |
1.75 |
1.78 |
1.67 |
1.75 |
1.45 |
17.97 |
17.47 |
25.46 |
17.83 |
23.11 |
–– |
–– |
–– |
–– |
–– |
||
|
N12 |
0.93 |
0.98 |
0.92 |
0.96 |
0.80 |
18.18 |
17.68 |
25.77 |
18.04 |
23.39 |
–– |
–– |
–– |
–– |
–– |
||
|
N13 |
1.22 |
1.24 |
1.16 |
1.22 |
1.01 |
–– |
–– |
–– |
–– |
–– |
–– |
–– |
–– |
–– |
–– |
||
|
N14 |
0.87 |
0.91 |
0.85 |
0.89 |
0.74 |
–– |
–– |
–– |
–– |
–– |
–– |
–– |
–– |
–– |
–– |
||
|
G1 |
1.75 |
1.78 |
1.67 |
1.75 |
1.45 |
19.86 |
19.32 |
28.16 |
19.71 |
25.56 |
–– |
–– |
–– |
–– |
–– |
||
|
G2 |
1.40 |
1.48 |
1.39 |
1.45 |
1.21 |
15.66 |
15.23 |
22.20 |
15.54 |
20.15 |
–– |
–– |
–– |
–– |
–– |
||
|
G3 |
1.29 |
1.31 |
1.23 |
1.28 |
1.07 |
13.54 |
13.17 |
19.20 |
13.44 |
17.42 |
–– |
–– |
–– |
–– |
–– |
||
|
G4 |
1.28 |
1.30 |
1.22 |
1.27 |
1.06 |
13.62 |
13.24 |
19.30 |
13.51 |
17.51 |
–– |
–– |
–– |
–– |
–– |
||
|
G5 |
1.37 |
1.39 |
1.30 |
1.36 |
1.14 |
–– |
–– |
–– |
–– |
–– |
–– |
–– |
–– |
–– |
–– |
||
|
G6 |
2.07 |
2.11 |
1.98 |
2.07 |
1.72 |
20.62 |
20.05 |
29.23 |
20.46 |
26.52 |
–– |
–– |
–– |
–– |
–– |
||
|
G7 |
2.35 |
2.39 |
2.24 |
2.34 |
1.95 |
9.92 |
9.87 |
14.39 |
10.07 |
13.06 |
–– |
–– |
–– |
–– |
–– |
||
|
G8 |
1.19 |
1.25 |
1.17 |
1.23 |
1.02 |
8.63 |
8.59 |
12.52 |
8.77 |
11.36 |
–– |
–– |
–– |
–– |
–– |
||
|
G9 |
1.97 |
2.01 |
1.89 |
1.97 |
1.64 |
7.30 |
7.26 |
10.58 |
7.41 |
9.60 |
–– |
–– |
–– |
–– |
–– |
||
|
G10 |
1.28 |
1.30 |
1.22 |
1.27 |
1.06 |
7.97 |
7.93 |
11.56 |
8.09 |
10.49 |
5.11 |
4.98 |
5.59 |
5.53 |
5.53 |
||
|
G11 |
1.32 |
1.35 |
1.27 |
1.32 |
1.10 |
13.11 |
13.05 |
19.02 |
13.32 |
17.26 |
–– |
–– |
–– |
–– |
–– |
||
|
Q1 |
2.94 |
3.03 |
2.84 |
2.97 |
2.48 |
26.76 |
26.64 |
38.83 |
27.18 |
35.24 |
6.08 |
5.93 |
6.65 |
6.59 |
6.59 |
||
|
Q2 |
4.05 |
4.17 |
3.91 |
4.09 |
3.41 |
16.56 |
16.48 |
24.02 |
16.82 |
21.80 |
6.11 |
5.96 |
6.69 |
6.62 |
6.62 |
||
|
Q3 |
2.50 |
2.58 |
2.42 |
2.53 |
2.11 |
16.19 |
16.11 |
23.48 |
16.44 |
21.31 |
7.21 |
7.03 |
7.89 |
7.81 |
7.81 |
||
|
Q4 |
2.57 |
2.65 |
2.49 |
2.60 |
2.17 |
27.30 |
26.55 |
38.70 |
27.09 |
35.12 |
6.78 |
6.61 |
7.42 |
7.34 |
7.34 |
||
|
Q5 |
2.81 |
2.89 |
2.71 |
2.83 |
2.36 |
23.16 |
22.52 |
32.83 |
22.98 |
29.79 |
8.04 |
7.84 |
8.80 |
8.71 |
8.71 |
||
|
Q6 |
3.84 |
3.95 |
3.71 |
3.87 |
3.23 |
21.86 |
21.26 |
30.99 |
21.69 |
28.12 |
7.81 |
7.62 |
8.55 |
8.47 |
8.47 |
||
|
Q7 |
2.54 |
2.62 |
2.46 |
2.57 |
2.14 |
22.01 |
21.40 |
31.19 |
21.84 |
28.31 |
5.62 |
5.48 |
6.15 |
6.09 |
6.09 |
||
|
Q8 |
3.04 |
3.13 |
2.94 |
3.07 |
2.56 |
28.17 |
27.40 |
39.94 |
27.96 |
36.24 |
6.97 |
6.80 |
7.63 |
7.56 |
7.56 |
||
|
Q9 |
3.08 |
3.17 |
2.97 |
3.11 |
2.59 |
28.06 |
27.29 |
39.78 |
27.85 |
36.10 |
6.73 |
6.56 |
7.36 |
7.29 |
7.29 |
||
|
Q10 |
2.74 |
2.82 |
2.65 |
2.76 |
2.30 |
16.11 |
15.67 |
22.84 |
15.99 |
20.73 |
6.89 |
6.72 |
7.54 |
7.47 |
7.47 |
||
QAMS1: trans-β-Apo-8’-carotenal as standard reference by using peak area to obtain RCF; QAMS2: trans-β-Apo-8’-carotenal as standard reference by using slope to obtain RCF; QAMS3: β-carotenal as standard reference by using peak area to obtain RCF; QAMS4: β-carotenal as standard reference by using slope to obtain RCF.
Table 3 The SMD between different QAMS methods with ESM (%) respectively
|
sample |
zeaxanthin dipalmitate |
zeaxanthin |
β-carotene |
||||||||||||
|
SMD1 |
SMD2 |
SMD3 |
SMD4 |
SMD1 |
SMD2 |
SMD3 |
SMD4 |
SMD1 |
SMD2 |
SMD3 |
SMD4 |
|
|||
|
N1 |
5.7 |
0.8 |
3.6 |
13.6 |
0.5 |
45.1 |
1.5 |
31.6 |
–– |
–– |
–– |
–– |
|
||
|
N2 |
1.8 |
4.5 |
0.2 |
16.9 |
1.6 |
48.1 |
3.7 |
34.4 |
–– |
–– |
–– |
–– |
|
||
|
N3 |
1.9 |
4.4 |
0.1 |
16.7 |
0.4 |
45.1 |
1.6 |
31.7 |
–– |
–– |
–– |
–– |
|
||
|
N4 |
6.2 |
0.4 |
4.1 |
13.3 |
–– |
–– |
–– |
–– |
–– |
–– |
–– |
–– |
|
||
|
N5 |
1.7 |
4.6 |
0.3 |
16.9 |
0.4 |
45.1 |
1.6 |
31.7 |
–– |
–– |
–– |
–– |
|
||
|
N6 |
1.7 |
4.6 |
0.3 |
16.9 |
2.7 |
41.8 |
0.7 |
28.7 |
–– |
–– |
–– |
–– |
|
||
|
N7 |
1.4 |
4.9 |
0.6 |
17.2 |
0.5 |
45.1 |
1.6 |
31.7 |
–– |
–– |
–– |
–– |
|
||
|
N8 |
1.7 |
4.5 |
0.3 |
16.9 |
2.7 |
41.8 |
0.7 |
28.7 |
–– |
–– |
–– |
–– |
|
||
|
N9 |
2.1 |
4.2 |
0.1 |
16.6 |
0.4 |
45.2 |
1.6 |
31.7 |
–– |
–– |
–– |
–– |
|
||
|
N10 |
1.6 |
4.7 |
0.4 |
17.0 |
2.8 |
41.7 |
0.8 |
28.6 |
–– |
–– |
–– |
–– |
|
||
|
N11 |
1.7 |
4.5 |
0.3 |
16.9 |
2.8 |
41.7 |
0.8 |
28.6 |
–– |
–– |
–– |
–– |
|
||
|
N12 |
5.4 |
1.1 |
3.3 |
13.9 |
2.8 |
41.8 |
0.8 |
28.6 |
–– |
–– |
–– |
–– |
|
||
|
N13 |
1.6 |
4.6 |
0.4 |
17.0 |
–– |
–– |
–– |
–– |
–– |
–– |
–– |
–– |
|
||
|
N14 |
4.6 |
1.8 |
2.5 |
14.5 |
–– |
–– |
–– |
–– |
–– |
–– |
–– |
–– |
|
||
|
G1 |
1.7 |
4.5 |
0.3 |
16.9 |
2.7 |
41.8 |
0.7 |
28.7 |
–– |
–– |
–– |
–– |
|
||
|
G2 |
5.7 |
0.8 |
3.6 |
13.6 |
2.7 |
41.8 |
0.8 |
28.6 |
–– |
–– |
–– |
–– |
|
||
|
G3 |
1.6 |
4.7 |
0.4 |
17.0 |
2.7 |
41.8 |
0.7 |
28.7 |
–– |
–– |
–– |
–– |
|
||
|
G4 |
1.6 |
4.7 |
0.4 |
17.0 |
2.8 |
41.7 |
0.8 |
28.6 |
–– |
–– |
–– |
–– |
|
||
|
G5 |
1.5 |
4.8 |
0.5 |
17.1 |
–– |
–– |
–– |
–– |
–– |
–– |
–– |
–– |
|
||
|
G6 |
1.9 |
4.3 |
0.1 |
16.7 |
2.8 |
41.7 |
0.8 |
28.6 |
–– |
–– |
–– |
–– |
|
||
|
G7 |
1.7 |
4.6 |
0.3 |
16.9 |
0.5 |
45.0 |
1.5 |
31.6 |
–– |
–– |
–– |
–– |
|
||
|
G8 |
5.0 |
1.4 |
3.0 |
14.2 |
0.5 |
45.1 |
1.6 |
31.7 |
–– |
–– |
–– |
–– |
|
||
|
G9 |
2.0 |
4.2 |
0.0 |
16.6 |
0.5 |
45.0 |
1.5 |
31.6 |
–– |
–– |
–– |
–– |
|
||
|
G10 |
1.6 |
4.7 |
0.4 |
17.0 |
0.5 |
45.0 |
1.5 |
31.6 |
2.5 |
9.3 |
8.3 |
8.3 |
|
||
|
G11 |
2.3 |
4.0 |
0.3 |
16.4 |
0.5 |
45.1 |
1.6 |
31.7 |
–– |
–– |
–– |
–– |
|
||
|
Q1 |
3.1 |
3.3 |
1.0 |
15.8 |
0.4 |
45.1 |
1.6 |
31.7 |
2.5 |
9.4 |
8.4 |
8.4 |
|
||
|
Q2 |
3.0 |
3.4 |
0.9 |
15.9 |
0.5 |
45.1 |
1.5 |
31.6 |
2.5 |
9.4 |
8.4 |
8.4 |
|
||
|
Q3 |
3.2 |
3.2 |
1.2 |
15.7 |
0.5 |
45.0 |
1.5 |
31.6 |
2.5 |
9.4 |
8.3 |
8.3 |
|
||
|
Q4 |
3.1 |
3.2 |
1.1 |
15.8 |
2.7 |
41.8 |
0.8 |
28.6 |
2.5 |
9.4 |
8.3 |
8.3 |
|
||
|
Q5 |
2.8 |
3.5 |
0.8 |
16.0 |
2.8 |
41.7 |
0.8 |
28.6 |
2.5 |
9.4 |
8.3 |
8.3 |
|
||
|
Q6 |
2.9 |
3.5 |
0.8 |
16.0 |
2.7 |
41.8 |
0.8 |
28.6 |
2.4 |
9.5 |
8.4 |
8.4 |
|
||
|
Q7 |
3.1 |
3.2 |
1.1 |
15.7 |
2.8 |
41.7 |
0.8 |
28.6 |
2.5 |
9.4 |
8.3 |
8.3 |
|
||
|
Q8 |
3.0 |
3.4 |
0.9 |
15.9 |
2.7 |
41.8 |
0.7 |
28.7 |
2.4 |
9.4 |
8.4 |
8.4 |
|
||
|
Q9 |
2.9 |
3.4 |
0.9 |
15.9 |
2.7 |
41.8 |
0.8 |
28.6 |
2.5 |
9.4 |
8.3 |
8.3 |
|
||
|
Q10 |
2.9 |
3.4 |
0.9 |
15.9 |
2.7 |
41.8 |
0.7 |
28.7 |
2.5 |
9.4 |
8.4 |
8.4 |
|
||
SMD1: SMD between ESM and QAMS 1 (trans-β-Apo-8’-carotenal as standard reference by using peak area to obtain RCF); SMD2: SMD between ESM and QAMS 2 (trans-β-Apo-8’-carotenal as standard reference by using slope to obtain RCF); SMD3: SMD between ESM and QAMS 3 (β-carotenal as standard reference by using peak area to obtain RCF); SMD4: SMD between ESM and QAMS 4 (β-carotenal as standard reference by using slope to obtain RCF).
4.- In most cases “content” should be replaced by “concentration”
Response: We checked our initial manuscript and changed “content” to “concentration” in some cases according to the meaning we want to express.
5.- If chromatographic separation has been carried out with a C30 column, all the references (including ruggedness study) to C18 columns should be deleted, since they are not relevant. In addition, 50 min for the separation of only 5 compounds is a very long time. An appropriate selection of solvents/temperature could greatly improve the analysis time.
Response: Thank you for your warm and insightful comments. We deleted the information of C18 in ruggedness tests and complemented the information of C30 in our revised manuscript. Actually, as you said, 50 min for the separation of only 5 compounds is a very long time. However, lutein and zeaxanthin are a pair of isomers, unless a smaller particle size column was used, it was difficult to achieve baseline separation only by optimizing the elution solvents. Smaller particle size of the column packing material was used, higher column pressure was obtained. Once the column temperature exceed than 25℃, lutein and zeaxanthin cannot be separated. All of the above factors led to a long time for the separation of 5 compounds in the manuscript.
6.- In line 117 “sample solution (sample N5 from Ningxia)” is mentioned, but it is the same than the rest of “sample solution” which appear in different parts of the manuscript. This must be clearly specified in all the cases, if it corresponds to a natural sample or a synthetic one.
Response: Thank you for your careful suggestion. In order to avoid this misconception, we deleted the sample information as the same batch of LB sample was used in most items of the methodological validation.
7.- In line 147 “a wide range of concentrations” in most cases do not reach 2 magnitude orders.
Response: Thank you for your careful suggestion. We revised the description in the revised manuscript as follows: All standard substances showed good linearity within the tested range of concentrations……
8.- In table 3, the intercept seems to be very high, and can be very relevant at low concentrations. In addition, the lower value of the range of lutein (5.66) should be 1.14 (LOQ).
Response: Thank you for your insightful comments. In our initial manuscript, the concentration range of the standard calibration curve took the LOQ as the lowest concentration, and the highest concentration was about 100 times LOQ. Based on the above principles, the calibration curve for each standard substance was established. Maybe our method could be improved, such as the selection of the highest concentration should be re-considered. Meanwhile, we corrected the lower value of the range of lutein in the revised manuscript.
9.- In line 158, ranges for recoveries are reported, but in table 4 only one value is given, and do not correspond to the previous ones. Please, revise. Values reported in line 172 do not appear in the table 4.
Response: We apologize for our careless mistakes. We now corrected them in the revised manuscript.
10.- In line 180, “low RSD value” should be “high RSD value”
Response: We now corrected it in the revised manuscript.
11.- What mean exactly the values reported in table 5? In addition, it is unclear the rest of conditions used for each experiment, e.g. when testing different instruments, what column/flow/temperature was chosen?
Response: Table 5 was the results of ruggedness tests. According to the guidelines of ChP2020, ruggedness refers to the degree to which the measurements results will not be affected when the chromatographic conditions have small changes, such as changes in LC instruments, columns, flow rates, column temperatures, etc. The chromatographic conditions in the manuscript was in section “3.2. Instruments and chromatographic conditions” as follows: HPLC-DAD analyses were performed on an Agilent 1260 Infinity HPLC system (equipped with 1260 Quat Pump VL, 1260 Vialsampler and 1260 DAD WR) by using a column of C30 (YMC, 4.6×250 mm, 5μm). A gradient of mobile phase was used for ef-ficient separation, mobile phases A (dichloromethane) and B (methane: acetonitrile: water, 81:14:5, v/v/v), with the elution program was as follows: 0-20 min, 30% A; 20-48 min, 50% A; 48-50 min, 70% A; 50-55 min, 70% A. The flow rate was 1 mL/min with the column temperature maintained at 22°C. Detection wavelength was 450 nm and the sample injection volume was 20 µL. When one of the conditions changes, the others remain unchanged. For example, when testing different instruments, the column is YMC C30, 4.6×250 mm, 5μm, the flow is 1 mL/min, and the column temperature is 22°C.
12.- Lines 201-203. The colour changed due to the storing period, but the coordinates (e.g. RGB) are not reported. Maybe it could be also a variable to be considered in statistical tests.
Response: Thank you for your insightful comments. The photos were taken during the storage of Lycium barbarum, but the clarity was not enough, so they were not provided in our manuscript. We considered that it have little effect on our data, but your warmly reminder will make our future experiments more rigorous. Thank you again.
13.- In line 213, no significant differences were observed, according to the authors, from SMD. What is the critical value? The underlined values in table 7 simply correspond to the highest value of the column?
Response: SMD >10% was used as a cutoff. In order to display the differences intuitively, underline indicated the highest value of SMD.
14.- In line 231, “Delinha” is really “Delingha”. Please, change also “was latitude” to “whose latitude”.
Response: We apologize for our careless mistakes. We now corrected them in the revised manuscript.
15.- Line 298: please, replace “methane” to “methanol”
Response: We now corrected it in the revised manuscript.
16.- Lines 305-306: Both dichloromethane and ethanol are very volatile solvents. Once prepared the standard solutions, the volume/mass was checked regularly along time to control unnoticed evaporation of solvent, thus increasing the concentration of solution?
Response: Thank you for your insightful and careful comments. In our actual experiments, we found dichloromethane is very easy to volatilize. In order to avoid the increasing the concentration of solution, the standard solutions were freshly prepared and the flasks were sealed with para-film and stored at 4℃. We complemented the corresponding descriptions in our revised manuscript as follows: Standard solutions were freshly prepared by dissolving approximately 10 mg of each standard substance, including trans-β-Apo-8’-carotenal, lutein, zeaxanthin, β-carotene, and zeaxanthin dipalmitate, into a 10 mL brown volumetric flask respectively, with dichloromethane as the solvent; and then diluted with anhydrous ethanol which contained 0.1% BHT (1000-fold dilution) to obtain different solutions of each standard for the determination of RCF using HPLC method. To prevent the volatilization of dichloromethane, the flasks were sealed with para-film and stored at 4℃.
17.- In section 3.4 the relative retention time is not described, as it could be expected from the title.
Response: We complemented the calculation of relative retention time in our revised manuscript as follows: The relative retention time (RRT) was calculated followed by equation (2).
(2)
tx and ts were the retention times of the analyte and standard reference to be measured in the standard solution, respectively.
18.- Line 355: High humidity was applied at high temperature? Ambient temperature? Both?
Response: Thank you for your careful comments. High humidity was applied at ambient temperature. In order to avoid this misconception, we revised the corresponding description as follows: …… the 34 batches of LB were stored in different stability conditions including high humidity (RH 75±5%) with ambient temperature (25±5℃), high temperature (45±5℃) with normal humidity (50±5%), and ambient temperature (25±5℃) with normal humidity (50±5%), respectively.
All the changes we made in the manuscript were highlighted in red colored text.

Reviewer 3 Report
The authors did a good work from an experimental point of view and I recommend the article for publication after some minor revisions.
More specific:
L19: The isomer ‘’trans’’ in italics and elsewhere mentioned in the text.
L40: Double bracket. Remove one.
L298: Methane or methanol? What kind of water?
L301: Which detector did you use? Model and company.
L310: What were the particle size of powders? Have you used a blender before?
L312: What was the ambient temperature? Was it static extraction?
Author Response
The authors did a good work from an experimental point of view and I recommend the article for publication after some minor revisions.
More specific:
L19: The isomer ‘’trans’’ in italics and elsewhere mentioned in the text.
Response: Thank you for your insightful comment. We revised isomer “trans” in italics in all of our revised manuscript.
L40: Double bracket. Remove one.
Response: We apologize for our careless mistakes. We now corrected it in the revised manuscript.
L298: Methane or methanol? What kind of water?
Response: It was methanol, we have corrected it in revised manuscript and complemented the origin of the water.
L301: Which detector did you use? Model and company.
Response: DAD was used in HPLC analyses, and we complemented the following instrument information in our revised manuscript: HPLC-DAD analyses were performed on an Agilent 1260 Infinity HPLC system (equipped with 1260 Quat Pump VL, 1260 Vialsampler and 1260 DAD WR)……
L310: What were the particle size of powders? Have you used a blender before?
Response: LB powders were sieved through a 80-mesh sieve with particle size was about 800 µm. We used a blender to mix the powders thoroughly. We revised the corresponding description as follows: LB samples were first dried under 60°C condition for 12 hours, powered and sieved through a 80-mesh sieve after cooling to ambient temperature (25±5°C). After mixing with a blender, approximately 1 g of LB powders were accurately weighted…….
L312: What was the ambient temperature? Was it static extraction?
Response: The ambient temperature was 25±5℃, we used an Elmasonic P 120H ultrasonic bath to prepare the sample solutions. We revised the corresponding description as follows: …….., extracted using an Elmasonic P 120H ultrasonic bath (Elma, Germany) for 30 min at ambient temperature (25±5℃), followed by filtered ……..
All the changes we made in the manuscript were highlighted in red colored text.

Round 2
Reviewer 2 Report
After a careful revision of the introduced changes, the manuscript can be accepted for publication in "Molecules".